# Effects of Timing of Acute and Consecutive Catechin Ingestion on Postprandial Glucose Metabolism in Mice and Humans

**DOI:** 10.3390/nu12020565

**Published:** 2020-02-21

**Authors:** Masaki Takahashi, Mamiho Ozaki, Miku Tsubosaka, Hyeon-Ki Kim, Hiroyuki Sasaki, Yuji Matsui, Masanobu Hibi, Noriko Osaki, Masashi Miyashita, Shigenobu Shibata

**Affiliations:** 1Organization for University Research Initiatives, Waseda University, 2-2 Wakamatsu-cho, Shinjuku, Tokyo 1628480, Japan; 2Graduate School of Advanced Science and Engineering, Waseda University, 2-2 Wakamatsu-cho Shinjuku, Tokyo 1628480, Japan; 3Faculty of Science and Engineering, Waseda Univesity, 2-2 Wakamatsu-cho, Shinjuku, Tokyo 1628480, Japan; 4AIST-National Institute of Advanced Industrial Science and Technology, Waseda University Computational Bio Big-Data Open Innovation Laboratory (CBBD-OIL), Tokyo 1690072, Japan; 5R&D, Core Technology, Biological Science Research, Kao Corporation, 2-1-3 Bunka Sumida, Tokyo 1318501, Japan; 6Faculty of Sport Sciences, Waseda University, 2-579-15 Mikajima Tokorozawa, Saitama 3591192, Japan

**Keywords:** glucose metabolism, epigallocatechin gallate, green tea, insulin, timing

## Abstract

We examined the effects of the timing of acute and consecutive epigallocatechin gallate (EGCG) and catechin-rich green tea ingestion on postprandial glucose in mice and human adults. In mouse experiments, we compared the effects of EGCG administration early (morning) and late (evening) in the active period on postprandial glucose. In human experiments, participants were randomly assigned to the morning-placebo (MP, *n* = 10), morning-green tea (MGT, *n* = 10), evening-placebo (EP, *n* = 9), and evening-green tea (EGT, *n* = 9) groups, and consumed either catechin-rich green tea or a placebo beverage for 1 week. At baseline and after 1 week, participants consumed their designated beverages with breakfast (MP and MGT) or supper (EP and EGT). Venous blood samples were collected in the fasted state and 30, 60, 120, and 180 min after each meal. Consecutive administration of EGCG in the evening, but not in the morning, reduced postprandial glucose at 30 (*p* = 0.006) and 60 (*p* = 0.037) min in the evening trials in mice. In humans, ingestion of catechin-rich green tea in the evening decreased postprandial glucose (three-factor analysis of variance, *p* < 0.05). Thus, catechin intake in the evening more effectively suppressed elevation of postprandial glucose.

## 1. Introduction

Postprandial hyperglycemia is related to the development of diabetes and cardiovascular disease [1,2,3]. Some studies have shown that glucose metabolism is influenced not only by meal size and energy composition but also by meal timing [4,5]. In fact, elevated postprandial glucose concentrations are higher in the evening than in the morning [6,7,8]. Moreover, the time-of-day variations in glucose tolerance, including insulin function, have been observed, peaking in the morning and with a minimum in the evening/night. The mechanisms underlying the time-of-day variations in glucose tolerance are caused by the circadian system, which also mediates time-of-day variations in digestion, absorption, and metabolism in the stomach and intestines [9,10]. Several transporters related to the absorption of glucose, including sodium/glucose cotransporter 1 (*SGLT1*), glucose transporter 2 (*GLUT2*), and *GLUT5*, have been shown to be expressed at higher levels during the active period compared with those during the inactive period in rodents [9,10]. Although several factors influence postprandial glucose metabolism, the effects of timing of antidiabetic food or beverage intake on postprandial glucose metabolism are still unclear.

Epigallocatechin-3-gallate (EGCG) is the most abundant catechin found in green tea, which possesses antidiabetic and anti-obesity properties [11,12]. Previous studies have demonstrated that regular green tea consumption is associated with reduced risk of diabetes [13,14]. In addition, the intake of green tea decreases fasting and postprandial glucose [15,16,17]. Recently, one study reported that acute ingestion of catechin-rich green tea at a different time of the day (i.e., morning or evening) exhibited varying glucose-lowering effects in humans [18]. Particularly, acute ingestion of catechin-rich green tea in the evening, but not in the morning, reduced postprandial glucose concentrations. Thus, the effects of green tea catechin intake on postprandial glucose concentrations differ according to variations in catechin absorption and metabolism controlled by the circadian rhythm. In addition, other studies have reported that consecutive green tea ingestion has additive effects on health-related parameters, which could contribute to the prevention of obesity and diabetes [19,20]. However, no studies have examined the consecutive effects of the timing of green tea ingestion on glucose and insulin concentrations. Furthermore, some recent studies have shown that short term (from 4 to 6 days) dietary interventions could change glucose metabolism and circadian rhythm in humans [21,22]. In addition, it has been shown that consecutive intake of caffeine or catechin could modify the peripheral circadian rhythm in mice [23,24]. Thus, these findings may be different from the effectiveness of the timing of consecutive EGCG and green tea intake with a short period (i.e., 1 week) on lowering glucose concentrations.

In the present study, we aimed to examine the effects of the timing of acute and consecutive EGCG and catechin-rich green tea ingestion on postprandial glucose and insulin concentrations in mice and young adults. We also compared the changes in postprandial catechin concentrations between morning and evening in humans. We hypothesized that the effectiveness of consecutive EGCG and green tea intake on lowering glucose and insulin concentrations may be greater in the evening than in the morning because elevated postprandial glucose concentrations are higher in the evening than in the morning.

## 2. Materials and Methods

### 2.1. Animal Experiments

#### 2.1.1. Animals and Materials

All animal experiments were performed and approved in accordance with the guidelines of the committee for Animal Experimentation at Waseda University (Permission #2017-A083). Eight-week-old male mice were purchased from Tokyo Laboratory Animals Science (Tokyo, Japan). All mice were housed in an animal room under standard conditions of relative humidity (60% ± 5%), temperature (22 ± 2 °C), and 12-h light-dark cycle (lights-on from 08:00 to 20:00). Zeitgeber time (ZT) 0 and ZT12 were designated as lights-on and lights-off times. The light intensity at the surface of the cage was approximately 100 lux. ICR mice were fed with a normal diet (EF; Oriental Yeast Co. Ltd., Japan) and water ad libitum before each experiment. EGCG was purchased from Bio Verde Co. Ltd. (Kyoto, Japan).

#### 2.1.2. Acute Treatment Protocol

The acute treatment protocol is shown in Figure 1. The ZT12 trial was defined as the morning trial, whereas the ZT20 trial was defined as the evening trial. On the day of blood glucose monitoring, all mice were randomly divided into four groups based on body mass and fasting blood glucose concentrations: morning-placebo group (*n* = 5), morning-EGCG group (*n* = 6), evening-placebo group (*n* = 5), and evening-EGCG group (*n* = 5). The sample size (*n* = 5–6 in each group) was determined based on a previous study [25]. In morning trials, the mice were administered water or EGCG (100 mg/kg) dissolved in water and corn starch (CS: 2 g/kg body weight [bw], i.g.) after overnight fasting and then underwent 2 h of blood glucose monitoring. In evening trials, the mice were administered water or EGCG (100 mg/kg) dissolved in water and CS (2 g/kg bw, i.g.) after 4 h fasting (ZT16–20) and then underwent 2 h of blood glucose monitoring. A previous study reported that this amount of EGCG resulted in decreased postprandial glucose concentrations in mice [25].

#### 2.1.3. Consecutive Treatment Schedule

The consecutive treatment schedule is shown in Figure 2. The ZT12 group was defined as the morning group, whereas the ZT20 group was defined as the evening group. All mice were divided into morning (*n* = 22) and evening (*n* = 22) groups. Mice in both groups were administered water or EGCG (100 mg/kg) dissolved in water at ZT12 or ZT20 for 4 days. Then, the mice were further divided into four groups: morning trial-placebo group (*n* = 5), morning trial-EGCG group (*n* = 6), evening trial-placebo group (*n* = 5), and evening trial-EGCG group (*n* = 6). The sample size (*n* = 6 in each group) was determined based on a previous study [25]. In the morning trials, the mice were administered only CS (2 g/kg bw, i.g.) dissolved in water at ZT12 after overnight fasting and then underwent 2 h of blood glucose monitoring. In the evening trials, the mice were administered the same amount of CS at ZT20 after 4 h fasting (ZT16–20) and then underwent 2 h of blood glucose monitoring.

#### 2.1.4. Blood Glucose and Insulin Measurements

Blood glucose concentrations were determined in blood samples collected from the tail vein at 0, 15, 30, 60, and 120 min after CS administration using a Glucose PILOT kit (Aventir Biotech, LLC, Carlsbad, CA, USA). The change in glucose concentration was assessed by the incremental area under the curve (AUC), which was calculated by the trapezoidal rule. Blood insulin concentrations were measured at 0, 30, 60, and 120 min after CS administration using an Ultra-sensitive Mouse Insulin Enzyme-linked Immunosorbent Assay kit (Mercodia AB, Uppsala, Sweden).

#### 2.1.5. Statistical Analysis

Data are presented as mean ± standard error of the mean (SEM). Kolmogorov-Smirnov tests were used to check for the normality of distributions for all blood parameters. For the analysis of normally distributed parameters, unpaired Student’s t-tests were used to assess differences in data at the fasting state between the control and EGCG groups at each time point (i.e., morning and evening trials) and to compare the AUC values. Two-factor analysis of variance (ANOVA) was used to determine the effects of group (control or EGCG) and postprandial interval (0–120 min) on the concentrations of glucose and insulin at each time point (i.e., morning and evening trials). When significant interaction effects were detected, we used the Tukey method for post-hoc comparisons. For the analysis of abnormally distributed parameters, Mann-Whitney U-tests (control and EGCG groups) and Kruskal-Wallis H tests (for more than three time points) were used to compare the values. Results with *p* values less than 0.05 were considered significant. Data analysis was performed using Predictive Analysis Software, version 23.0 for Windows (SPSS Japan Inc., Tokyo, Japan).

### 2.2. Human Experiments

#### 2.2.1. Participants

Healthy young adults (*n* = 39; 18 men and 21 women) who met the following inclusion criteria were recruited for this study: (1) not using glucose/insulin-lowering or related medications, (2) no changes in body weight of more than 5% within 6 months, (3) not heavy drinkers (men: < 40 g/day, women: < 20 g/day), (4) not been diagnosed with diabetes or dyslipidaemia by a doctor, and (5) not taking any antioxidant, anti-obesity, or antidiabetes supplements. This study was conducted according to the guidelines of the Declaration of Helsinki and was approved by the ethics committees of Waseda University (approval number: 2017-016). Informed consent was obtained from all participants after the experiment was described to them in detail. Human trial of the present study is registered at www.umin.ac.jp/ctr/ as UMIN000027486.All participants completed a questionnaire on physical activity, exercise, dietary intake, lifestyle habits, and health status prior to the study. None of the study participants were trained athletes competing in any sporting event, but some were recreationally active. One participant was excluded from the study because she did not participate in the follow-up. Consequently, 38 participants were included in the analysis.

#### 2.2.2. Main Trial

A double-blind, placebo-controlled, parallel design was used. The participants were randomly assigned to the following groups: morning-placebo (MP, *n* = 10; men, *n* = 4; women, *n* = 6), morning-green tea (MGT, *n* = 10; men, *n* = 5; women, *n* = 5), evening-placebo (EP, *n* = 9; men, *n* = 4; women, *n* = 5), and evening-green tea (EGT, *n* = 9; men, *n* = 4; women, *n* = 5).

Each participant attended two laboratory-based tests at baseline and after 1 week. Participants in the MP and MGT groups were required to visit the laboratory at 0830 after a minimum 10-h overnight fast (no intake of food or drink, except water), whereas participants in the EP and EGT groups were required to visit the laboratory at 1630 after a 4-h fasting state (no intake of food or drink, except water). For all groups, a fasting venous blood sample was collected by venipuncture while the participants were in a seated position after 10–15 min. Further venous blood samples were collected at 30, 60, 120, and 180 min after providing the test meal in each trial.

#### 2.2.3. Intervention

After the first trial, the participants were required to consume test beverages daily from day 2 to day 6 at approximately the same time. Participants in the MP and MGT group were requested to consume the test beverage (i.e., placebo or catechin-rich green tea beverage) from 5:00 AM to 10:00 AM, whereas those in the EP and EGT groups were requested to the consume test beverage from 5:00 PM to 10:00 PM.

#### 2.2.4. Test Meals

The test meals were prepared according to the estimated average required energy per day for each participant. The energy of the test meal was distributed as follows: 15% from fat, 70% from carbohydrates, and 15% from protein. This percentage of carbohydrate loading could potentially increase postprandial glucose in healthy adults [26,27]. All participants were asked to consume the test meal within 20 min. The time taken to consume the meal during the first test was recorded and replicated in the second test (i.e., 1 week after the first test). None of the participants reported nausea or any gastrointestinal discomfort in all trials. The participants consumed a whole bottle of room temperature test beverage (i.e., 20–25 °C) during the first trial, and the pattern and volume ingested were replicated in the second trial.

#### 2.2.5. Green Tea and Placebo Beverage Contents

The green tea and placebo beverages used in this study were provided by Kao Corporation (Tokyo, Japan). The contents of catechin-rich green tea and placebo beverages are summarized in Table 1. We used a brewed green tea with a natural flavor as the base beverage, whereas the catechin-rich green tea was prepared using green tea extract and hot water. The green tea beverage contained 615 mg/350 mL total catechins (31 mg catechin, 22 mg catechin gallate, 120 mg gallocatechin, 97 mg gallocatechin gallate, 38 mg epicatechin, 45 mg epicatechin gallate, 127 mg epigallocatechin, and 135 mg EGCG) and 85 mg caffeine. Previous our studies reported that this specific amount of catechin (equivalent to 5–6 cups of green tea daily) resulted in decreased postprandial glucose concentrations in humans [17,18]. The placebo beverage contained 0 mg/350 mL total catechins and 80 mg caffeine. The catechin-rich green tea and placebo beverages were adjusted to contain the same levels of vitamin C and other minor polyphenols. Each beverage was also matched and adjusted based on appearance and flavor, such that they could not be distinguished.

#### 2.2.6. Standardization of the Diet and Physical Activity

All participants were requested to follow the same diet before each trial (i.e., supper for the morning trials and lunch for the evening trials) at both baseline and after 1 week. They were also requested to refrain from drinking alcohol the previous day for each trial. In addition, they were requested to stay inactive and to not perform high-intensity physical activity on the day before all trials and throughout the experimental period and to maintain their daily lifestyle (i.e., dietary habits, physical activity patterns, and sleep-awake cycle) during the intervention period.

#### 2.2.7. Measurements of Anthropometry and Chronotype

Anthropometric variables were measured at baseline and after the intervention. Body mass was measured to the nearest 0.1 kg using a digital balance (In Body; In Body Japan, Tokyo, Japan). Body mass index (BMI) was also calculated by In Body. Arterial blood pressure was measured from the left arm with the participant in a seated position using a sphygmomanometer (OMRON HEALTHCARE Co., Ltd., Kyoto, Japan). These measurements were obtained at baseline and after 1 week. Participants were seated in a chair for 5 min before obtaining the measurements.

Chronotype was assessed using the Horne–Ostberg Morningness-Eveningness Questionnaire (MEQ) [28], which consisted of 19 questions related to preferred sleep time and daily performance (i.e., what would be the best time to perform hard physical work?). The scores ranged from 16 to 86. From their scores, the participants were divided into three chronotype groups: morningness (score 59–86), intermediate (score 42–58), or eveningness (score 16–41).

#### 2.2.8. Blood Collection and Analysis

For plasma catechin measurements, venous blood samples were collected into tubes containing heparin-sodium. Plasma gallocatechin, epigallocatechin, epigallocatechin gallate, gallocatechin gallate, epicatechin gallate, and catechin gallate concentrations were measured using high-performance liquid chromatography with solid-phase extraction as previously described [29,30].

For plasma glucose measurements, venous blood samples were collected into tubes containing sodium fluoride-ethylenediaminetetraacetic acid. The samples were immediately centrifuged at 3000 rpm for 10 min at 4 °C, and the serum was dispensed into plain microtubes and stored at −80 °C until the assay. For serum insulin, venous blood samples were collected into tubes containing clotting activators for isolation of serum, and the samples were allowed to clot for 30 min at room temperature and centrifuged. Enzymatic colorimetric assays were performed to measure the plasma concentrations of glucose (GLU-HK (M); Shino-test Corporation, Kanagawa, Japan). Plasma concentrations of insulin (Mercodia Insulin ELISA; Mercodia AB, Uppsala, Sweden) were measured by enzyme-linked immunosorbent assay.

#### 2.2.9. Statistical Analysis

Data are presented as mean ± SEM. Kolmogorov-Smirnov tests were used to check the normality of distributions for all blood parameters. Based on the distribution of postprandial glucose values in our previous studies, the sample size was calculated to detect a large effect (Cohen’s d = 0.98) [31,32]. The calculated sample size of 11 was required to approximately have 80% power to detect large effects at a significance level of 0.05. One-factor ANOVA was used to compare baseline values for all groups, whereas three-factor ANOVA was used to determine the effects of the intake timing (morning or evening), intervention (baseline or after 1 week), and postprandial interval (0–180 min) on the concentrations of individual catechin. Two-factor ANOVA was used to evaluate the effects of the intake timing and intervention on AUC of all catechins in the plasma. In addition, three-factor repeated-measures ANOVA was used to determine the effects of the beverage (placebo or catechin-rich green tea), intervention (baseline or after 1 week), and postprandial interval (0–180 min) at each meal timing (morning or evening) on the concentrations of glucose and insulin. Two-factor ANOVA was used to evaluate the effects of the beverage and intervention at each meal timing on the AUC of glucose and insulin concentrations. When significant interaction effects were detected, we used the Bonferroni method for post-hoc comparisons. Results with *p* values less than 0.05 were considered significant. Data analysis was performed using Predictive Analysis Software, version 23.0 for Windows (SPSS Japan Inc. Tokyo, Japan).

## 3. Results

### 3.1. Animal Experiments

#### 3.1.1. Effects of Timing of Acute EGCG Administration on Blood Glucose Concentrations

Mann-Whitney tests showed that the concentrations of glucose in the EGCG group at fasting state (*p* = 0.006), 15 min (*p* = 0.006), 30 min (*p* = 0.006), and 60 min (*p* = 0.017) after CS administration were significantly lower than those in the control group for the morning trials. The AUC values of blood glucose were also significantly lower in the EGCG group in the morning trial (Figure 3A). In contrast, the concentrations of glucose in the EGCG group at 120 min (*p* = 0.045) after CS administration were significantly higher than those in the control group in the morning trials. Moreover, the concentrations of glucose in the EGCG group at 30 min (*p* = 0.016) after CS administration were significantly lower than those in the control group in the evening trials (Figure 3B). In contrast, the concentrations of glucose in the EGCG group at 120 min (*p* = 0.009) after CS administration were significantly higher than those in the control group for the evening trials. 

#### 3.1.2. Effects of Timing of Consecutive EGCG Administration on Blood Glucose and Insulin Concentrations

There were no significant differences in the concentrations of glucose in the morning group in both morning and evening trials. In the evening group, the concentrations of glucose in the EGCG group at 30 (*p* = 0.006) and 60 min (*p* = 0.037) after CS administration were significantly lower than those in the control group for the evening trials (Figure 4D).

In the morning group, the concentrations of insulin in the EGCG group in the fasting state (*p* = 0.028) and at 60 min (*p* = 0.009) after CS administration were significantly lower than those in the control group for the morning trials (Figure 5A). In contrast, in the evening group, the concentrations of insulin in the EGCG group at 30 (*p* = 0.028) and 60 min (*p* = 0.011) after CS administration were significantly higher than those in the control group for the evening trials (Figure 5D). In addition, the AUC of blood insulin was significantly higher in the EGCG group for the evening trials (*p* = 0.045) (Figure 5D).

### 3.2. Human Experiments

#### 3.2.1. Effects of Timing of Acute and Consecutive Catechin-Rich Green Tea Intake on Physical Characteristics

There were no significant differences in the physical characteristics at baseline for all groups (Table 2). There were also no significant differences in the beverage intake time and MEQ score among groups (Appendix A). In addition, the intervention did not affect the physical characteristics and MEQ score of the participants for all groups.

#### 3.2.2. Effects of Timing of Acute and Consecutive Catechin-Rich Green Tea Intake on Serum Catechin Concentrations

Plasma concentrations of gallocatechin, epigallocatechin, EGCG, gallocatechin gallate, epicatechin gallate, and catechin gallate are shown in Figure 6A–F. In the MP and EP groups, in which catechin-rich green tea was not consumed, blood catechins were barely detectable. At the baseline level, there were no differences in all individual catechins in the plasma among all groups. For plasma concentrations of gallocatechin (*p* = 0.001), epigallocatechin (*p* = 0.001), EGCG (*p* = 0.020), and gallocatechin gallate (*p* = 0.001), three-factor ANOVA showed a significant main effect of intake timing. Additionally, a significant main effect of intervention was found for plasma concentrations of gallocatechin (*p* = 0.042), EGCG (*p* = 0.001), gallocatechin gallate (*p* = 0.007), and epicatechin gallate (*p* = 0.034). For all individual catechins, except gallocatechin gallate, three-factor ANOVA showed a significant main effect of time (*p* = 0.001). In addition, a significant intake timing × time interaction was found for plasma concentrations of EGCG (*p* = 0.008), gallocatechin gallate (*p* = 0.034), and epicatechin gallate (*p* = 0.005). Post-hoc tests showed that epigallocatechin gallate concentration at 180 min (*p* = 0.025) after meal intake were significantly lower in the EGT group than in the MGT group at the baseline. Gallocatechin gallate at 60 min (*p* = 0.038), 120 min (*p* = 0.011), and 180 min (*p* = 0.003) after meal intake were significantly lower in the EGT group than in the MGT group at the baseline. In addition, gallocatechin gallate at 120 min (*p* = 0.038), and 180 min (*p* = 0.045) after meal intake were significantly lower in the EGT group than in the MGT group at the after 1 week. Epicatechin gallate at 30 min (*p* = 0.025) after meal intake were significantly higher in the EGT group than in the MGT group at the after 1 week.

The AUC values of all catechins in the plasma are shown in Appendix A. For the AUC values of gallocatechin (*p* = 0.002), epigallocatechin (*p* = 0.002), and gallocatechin gallate (*p* = 0.015), two-factor ANOVA showed a significant main effect of intake timing.

#### 3.2.3. Effects of Timing of Acute and Consecutive Catechin-Rich Green Tea Intake on Blood Glucose and Insulin Concentrations

Plasma concentrations of glucose are shown in Figure 7A,B. At the baseline level, there were no differences in plasma glucose concentrations among all groups. In the morning groups, three-factor ANOVA showed significant main effects of beverage (*p* = 0.049) and time (*p* = 0.001) and a significant beverage × time interaction (*p* = 0.030). Post-hoc tests showed that glucose concentration at 60 min (*p* = 0.049) after meal intake were significantly higher in the MGT group than in the MP group at the baseline. In the evening groups, three-factor ANOVA showed significant main effects of beverage (*p* = 0.001) and time (*p* = 0.001). The AUC values of plasma glucose concentrations are shown in Figure 7C,D). Although there were no significant effects of the AUC for glucose concentrations in the morning groups, two-factor ANOVA showed a significant main effect of beverage (*p* = 0.001) in the evening groups.

Plasma concentrations of insulin are shown in Figure 8A,B. At the baseline level, there were no differences in plasma insulin concentrations among all groups. In the morning groups, three-factor ANOVA showed significant main effects of time (*p* = 0.001) and a significant beverage × time interaction (*p* = 0.037). In the evening groups, three-factor ANOVA showed significant main effects of beverage (*p* = 0.001) and time (*p* = 0.001). The AUC values of plasma insulin concentrations are shown in Figure 8C,D. Although there were no significant effects on the AUC values of insulin concentrations in the morning groups, two-factor ANOVA showed a significant main effect of beverage (*p* = 0.013) in the evening groups.

## 4. Discussion

To the best of our knowledge, our study is the first to examine the effects of timing of consecutive catechin ingestion on postprandial glucose metabolism. Our main finding was that the consecutive administration of EGCG in the evening, but not in the morning, reduced postprandial glucose in mice. We also observed that acute and consecutive ingestion of catechin-rich green tea in the evening decreased postprandial glucose in humans. In contrast, the consecutive administration of EGCG and ingestion of catechin-rich green tea in the evening, but not in the morning, increased postprandial insulin in both mice and humans. Thus, our results indicated that intake of catechins, including EGCG, in the evening was more effective in suppressing the elevation of postprandial glucose.

Glucose tolerance becomes worse from morning to evening in humans. In fact, previous studies have reported that postprandial glucose concentrations are more strongly elevated following evening meals than after morning meals [6,7,8]. Our results were consistent with these previous studies because the concentrations of glucose in the evening groups at 120 min after consuming a meal were significantly higher than those in the morning groups (Appendix A). These responses may be attributed to decreased insulin sensitivity and beta-cell function and action in the evening. Moreover, some studies have shown that postprandial insulin concentrations at breakfast are higher than those at dinner [33,34,35]. Our results also showed that the concentrations of insulin in the morning groups after a meal were significantly higher than those in the evening groups (Appendix A). These results suggested that a meal in the evening exacerbated the state of postprandial glucose by decreasing insulin function. Therefore, it is important to consider the effectiveness of antidiabetic foods and beverages at different administration times, focusing on dinner time.

Green tea polyphenols, particularly catechins, have antidiabetic and anti-obesity functions [1,15,36]. These functions are attributed to the presence of polyphenols, such as EGCG, epigallocatechin, epicatechin gallate, and epicatechin. Although previous studies have measured blood catechin concentrations after drinking to evaluate of antidiabetic and antioxidant effects, only a few studies have compared the acute and consecutive effects of the timing of green tea ingestion on blood catechin concentrations after drinking green tea [17,37]. In the current study, the intake of catechin-rich green tea in the morning and evening groups resulted in a significant increase in plasma catechin concentrations, supporting our previous findings and those of others [17,37]. Notably, there were significant timing effects (i.e., morning or evening intake) on the concentrations of gallocatechin, epigallocatechin, EGCG, and gallocatechin gallate. One factor that may have caused these differences is the fasting period as the fasting period in the evening group was shorter than that in the morning group. This may have influenced the absorption of catechins in the stomach and small intestine. In addition, postprandial gallocatechin, EGCG, gallocatechin gallate, and epicatechin gallate concentrations were significantly elevated after intervention. These findings indicated that timing and the continuous intake of catechins were important factors for evaluating postprandial catechin concentrations.

In the current study, consecutive EGCG administration in the evening attenuated postprandial plasma glucose concentrations in mice. In addition, we showed that acute and consecutive catechin-rich green tea intake in the evening, but not in the morning, suppressed the elevation of postprandial glucose in humans. Although it is difficult to explain the mechanism through which catechin-rich green tea intake in the morning increases postprandial glucose, some studies have reported that a combination of green tea polyphenols and amylase can produce a high and sustained postprandial glycemic response [38,39]. Our test meal was composed mainly of white rice, which contains abundant amounts of amylose, thereby providing high carbohydrate content. Our results were also consistent with our previous study [18]. Importantly, catechin-rich green tea intake in the evening showed greater glucose-lowering effects than glucose-elevating effects owing to the combination of green tea polyphenols and amylose. Therefore, these results indicated that intake of catechin-rich green tea in the evening was more effective for improving glucose metabolism.

The mechanisms through which postprandial glucose levels are lowered by green tea intake could involve increased glucose uptake via glucose transporters or the activation of the insulin signal pathway. Green tea catechins have been reported to increase adipocyte insulin receptor binding and membrane GLUT4 protein content in rats [40,41]. Another study reported that EGCG increases the expression of genes associated with insulin sensitivity [42]. Moreover, EGCG and green tea ingestion may inhibit intestinal α-amylase, sucrose, and α-glucosidase, which reduce carbohydrate absorption in the intestine [43]. Therefore, green tea catechins have multiple biological activities that could exert antidiabetic effects. However, these activities may not be modulated by circadian rhythms. In fact, previous studies have shown that carbohydrate digesting enzymes (i.e., α-amylases and α-glucosidases) and glucose transporters (i.e., SGLT1 and GLUT2) have circadian rhythms [9,10,25,44]. These variations influence the effects of green tea intake on postprandial glucose responses with different meal timing.

Another mechanism affecting the glucose-lowering effects of green tea catechin intake is insulin response. Previous studies have shown that acute ingestion of green tea decreases postprandial insulin sensitivity in young and older adults [16,17]. In the current study, the administration of EGCG in morning trials in mice and the intake of catechin-rich green tea in the morning groups in humans decreased postprandial insulin. Notably, the administration of EGCG in evening trials in mice and the intake of catechin-rich green tea in evening trials in humans increased postprandial insulin. Considering the decreased insulin sensitivity in the evening, our results suggested that intake of EGCG or catechin-rich green tea may attenuate this decreased insulin sensitivity in the evening.

There are some limitations of our study. First, the fasting period was shorter in evening trials than in morning trials to mimic the daily lifestyle of most people. In general, the fasting period before breakfast is the longest. Therefore, the metabolic conditions in the morning and evening trials and groups were different. Second, both the glucose and insulin concentrations were not affected by the proposed interventions, which can be attributed to our short experimental period. In contrast, most studies investigated the effects of green tea catechins on fasting and postprandial glucose for a long period of time (ranging from 4 to 16 weeks) [36,45]. Thus, long-term research may be required to confirm the chronic effects of catechin intake at different times on fasting and postprandial glucose in humans.

## 5. Conclusions

We demonstrated that acute administration of EGCG and ingestion of catechin-rich green tea in the evening may be effective for lowering postprandial glucose through increasing postprandial insulin, but not through exerting consecutive effects on glucose and insulin functions in humans.

## Figures and Tables

**Figure 1 nutrients-12-00565-f001:**
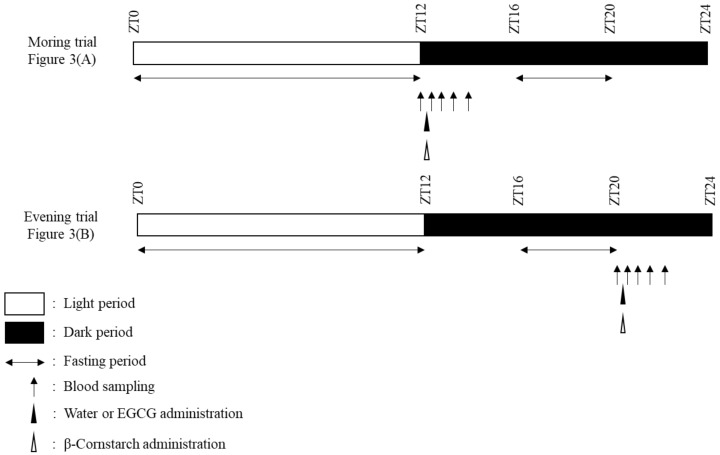
Protocol for acute experiments in mice.

**Figure 2 nutrients-12-00565-f002:**
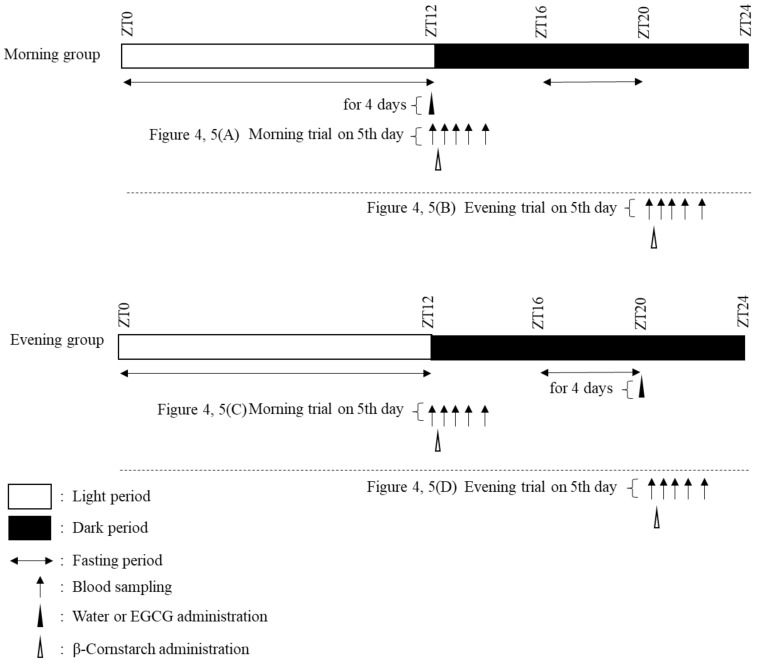
Protocol for consecutive experiments in mice.

**Figure 3 nutrients-12-00565-f003:**
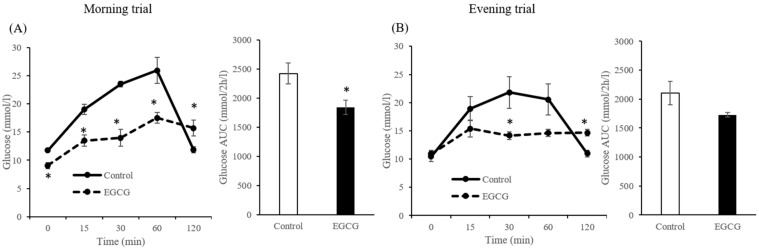
Concentrations and incremental area under the curve (AUC) of plasma glucose in the morning (**A**) and evening (**B**) trials in mice. Data are expressed as the mean and SEM, represented by bidirectional bars. * Mean values were significantly different from that of control group at same time.

**Figure 4 nutrients-12-00565-f004:**
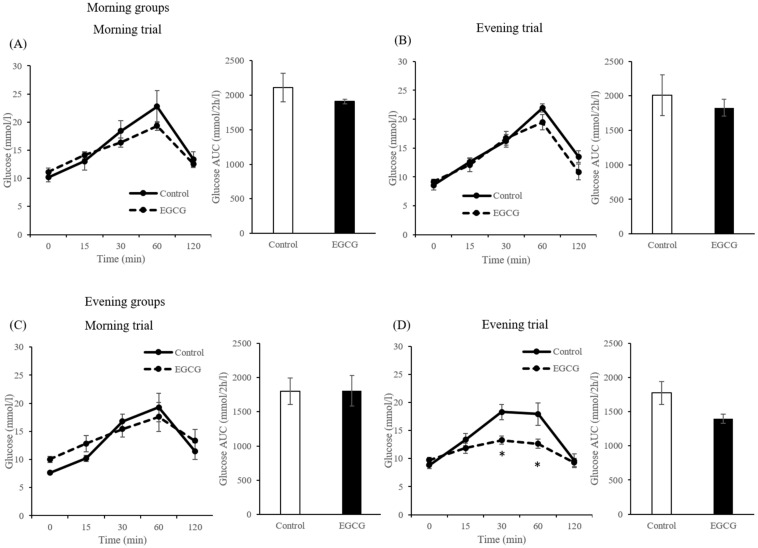
Concentrations and incremental area under the curve (AUC) of plasma glucose in the morning groups ((**A**): morning trials, (**B**): evening trials) and evening groups ((**C**): morning trials, (**D**): evening trials) in mice. Data are expressed as the mean and SEM, represented by bidirectional bars. * Mean values were significantly different from that of control group at same time.

**Figure 5 nutrients-12-00565-f005:**
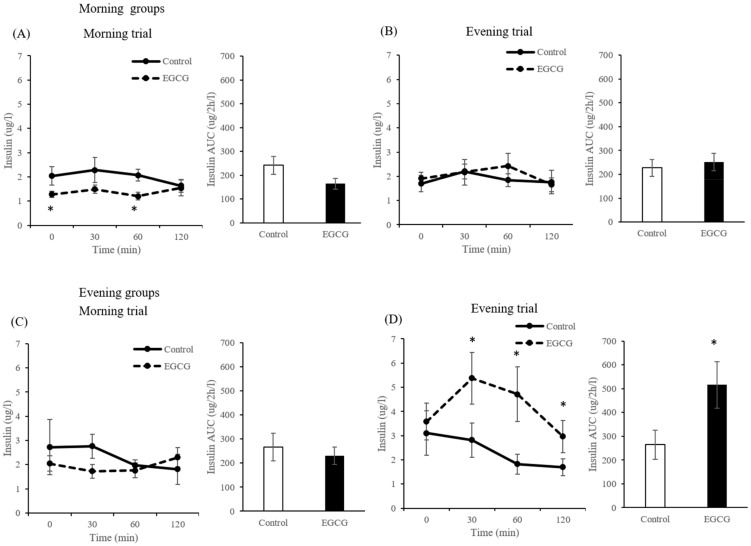
Concentrations and incremental area under the curve (AUC) of plasma insulin in the morning groups ((**A**): morning trials, (**B**): evening trials) and evening groups ((**C**): morning trials, (**D**): evening trials) in mice. Data are expressed as the mean and SEM, represented by bidirectional bars. * Mean values were significantly different from that of control group at same time.

**Figure 6 nutrients-12-00565-f006:**
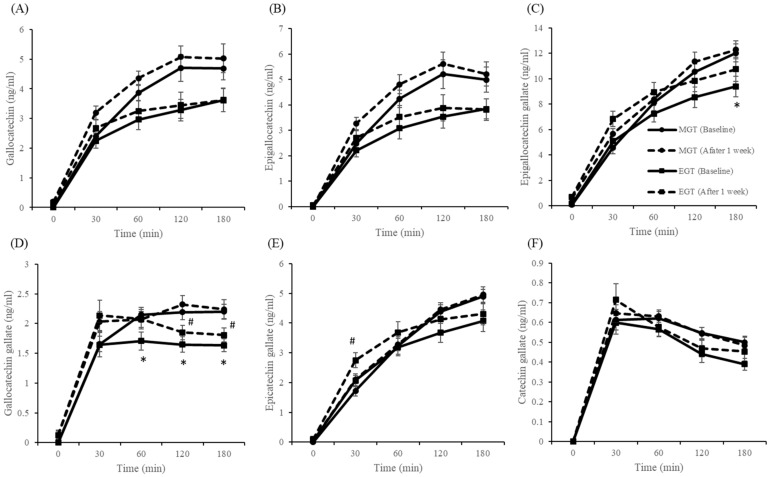
Fasting and postprandial plasma concentrations of gallocatechin (**A**), epigallocatechin (**B**), epigallocatechin gallate (**C**), gallocatechin gallate (**D**), epicatechin gallate (**E**), and catechin gallate (**F**). Data are expressed as the mean and SEM, represented by bidirectional bars. * Mean values were significantly different from that of MGT group at same meal time (baseline) (*p* < 0.05), # Mean values were significantly different from that of MGT group at same meal time (after 1 week) (*p* < 0.05).

**Figure 7 nutrients-12-00565-f007:**
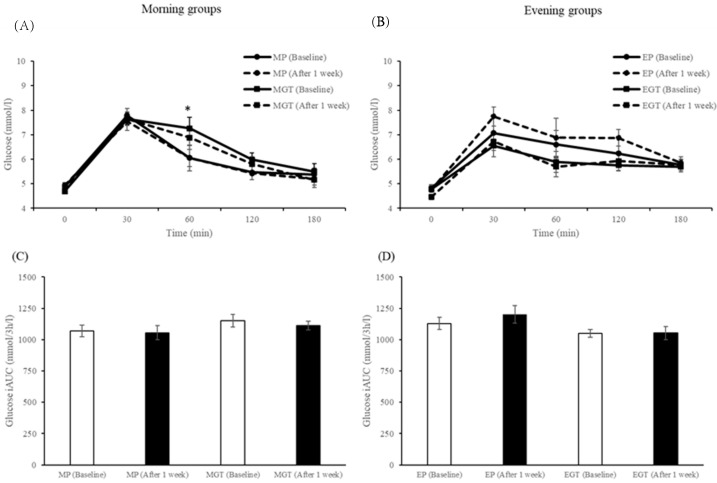
Concentrations and incremental area under the curve (AUC) of plasma glucose in the morning groups (**A**,**C**) and evening groups (**B**,**D**) in humans. Data are expressed as the mean and SEM, represented by bidirectional bars. * Mean values were significantly different from that of MP group at same meal time (baseline) (*p* < 0.05).

**Figure 8 nutrients-12-00565-f008:**
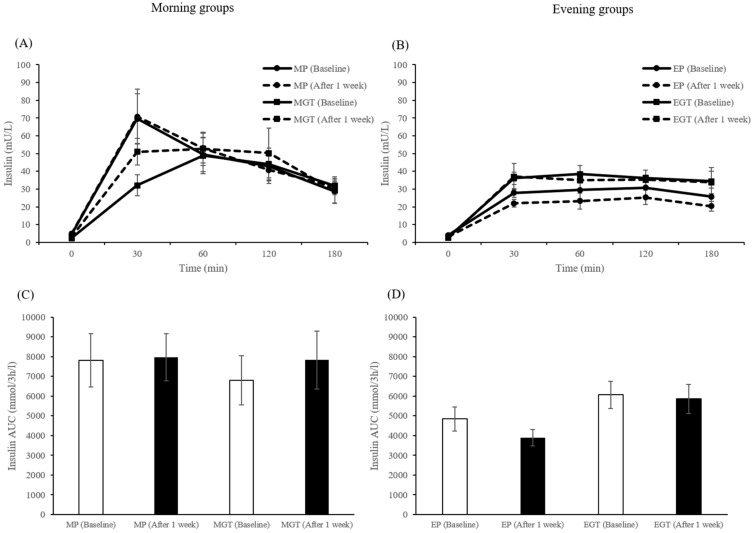
Concentrations and incremental area under the curve (AUC) of plasma insulin in the morning groups (**A**,**C**) and evening groups (**B**,**D**) in humans. Data are expressed as the mean and SEM, represented by bidirectional bars.

**Table 1 nutrients-12-00565-t001:** Contents of catechin-rich green tea (GT) and placebo beverage (P).

	GT	P
Water (g)	350	350
Energy (kcal)	18	0
Carbohydrate (g)	4	0
Fat (g)	0	0
Protein (g)	0	0
Catechin (mg)	31	0
Catechin gallate (mg)	22	0
Gallocatechin (mg)	120	0
Gallocatechin gallate (mg)	97	0
Epicatechin (mg)	38	0
Epicatechin gallate (mg)	45	0
Epigallocatechin (mg)	127	0
Epigallocatechin gallate (mg)	135	0
Caffeine (mg)	85	80

**Table 2 nutrients-12-00565-t002:** Physical characteristics at baseline and after 1 week.

		Group			
		MP (*n* = 10)	MGT (*n* = 10)	EP (*n* = 9)	EGT (*n* = 9)
Age (years)	Baseline	24.7 ± 1.4	23.3 ± 1.1	22.3 ± 0.7	24.4 ± 1.4
	After 1 week	24.7 ± 1.4	23.3 ± 1.1	22.3 ± 0.7	24.4 ± 1.4
Height (m)	Baseline	1.64 ± 0.03	1.65 ± 0.03	1.64 ± 0.03	1.67 ± 0.04
	After 1 week				
Body mass (kg)	Baseline	59.1 ± 3.1	57.5 ± 3.9	56.6 ± 3.5	60.3 ± 4.1
	After 1 week	58.8 ± 3.1	55.6 ± 2.4	56.5 ± 3.4	60.2 ± 4.1
Body mass index (kg/m^2^)	Baseline	21.8 ± 0.6	20.3 ± 0.5	20.8 ± 0.8	21.4 ± 0.6
	After 1 week	21.7 ± 0.6	20.4 ± 0.5	20.8 ± 0.8	21.4 ± 0.6
Systolic blood pressure (mmHg)	Baseline	114.6 ± 2.5	110.9 ± 2.5	114.5 ± 4.3	114.4 ± 4.4
	After 1 week	114.2 ± 2.6	111.7 ± 2.3	113.2 ± 4.9	113.6 ± 4.0
Diastolic blood pressure (mmHg)	Baseline	73.6 ± 2.9	74.6 ± 2.4	69.0 ± 3.6	68.2 ± 3.2
	After 1 week	70.1 ± 3.1	71.1 ± 1.5	69.1 ± 2.9	70.0 ± 2.4

MP, morning-placebo; MGT, morning-green tea; EP, evening-placebo; EGT, evening-green tea.

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
