# Peer review of "Effects of Timing of Acute and Consecutive Catechin Ingestion on Postprandial Glucose Metabolism in Mice and Humans"

_nutrients, 2020, doi:10.3390/nu12020565_

Round 1
Reviewer 1 Report
In this manuscript, the authors investigated the impact of the timing of catechin-rich green tea consumption on blood glucose level in both animal model and humans. The found that consuming green tea in the evening is more efficient to lower the blood glucose level.
Although the topic of this manuscript fits well with the scope of Nutrients, I am very concerned about the novelty of this study. “Nutrients will consider manuscripts for publication that provide novel insights into the impacts of nutrition on human health or novel methods for assessing nutritional status.” In 2019, the authors published a paper titled “Effects of timing of acute catechin-rich green tea ingestion on postprandial glucose metabolism in healthy men” in the Journal of Nutritional Biochemistry (reference #18). The conclusion from this previous study was very similar to the current manuscript. The condition of human study in both cases had only minor difference. I cannot see that the current manuscript provide much “novel insights into the impacts of nutrition on human health”.
I also have some concerns about the experimental part of the manuscript.
1. Figure 6 showed that placebo groups also has detectable level of catechin. This raises the question whether the quantification of catechin is reliable or not. The authors did not provide details about how these data were obtained in the manuscript.
2. In Figure 6E, the error bars are missing.
3. The data showing statistically significant difference should be marked with “*” in all the figures. Please check Figures 6-8 and supplementary Figures S1-S3.
4. The data collected in section 2.2.6 and 2.2.7 were not presented in the results section. Due to the limited number of participates in the human study, I wonder whether the authors can get statistically significant results from these data or not.
There are a few typos in the manuscripts as well, including “standardization” on line 188, “concentration” on line 113 and 229.
Author Response
Paper No. Nutrients- 692047 R1
Response to Reviewers’ Comments
Once again, we wish to thank the editor and reviewers for reading our manuscript so thoroughly and providing such constructive feedback. The quality of our manuscript has certainly improved as a result of these comments. Our point-by-point responses are provided below, and the necessary changes are highlighted in yellow in the revised manuscript.
Reviewer 1:
General comments
In this manuscript, the authors investigated the impact of the timing of catechin-rich green tea consumption on blood glucose level in both animal model and humans. The found that consuming green tea in the evening is more efficient to lower the blood glucose level.
Although the topic of this manuscript fits well with the scope of Nutrients, I am very concerned about the novelty of this study. “Nutrients will consider manuscripts for publication that provide novel insights into the impacts of nutrition on human health or novel methods for assessing nutritional status.” In 2019, the authors published a paper titled “Effects of timing of acute catechin-rich green tea ingestion on postprandial glucose metabolism in healthy men” in the Journal of Nutritional Biochemistry (reference #18). The conclusion from this previous study was very similar to the current manuscript. The condition of human study in both cases had only minor difference. I cannot see that the current manuscript provide much “novel insights into the impacts of nutrition on human health”.
Response: According to your suggestion, we have revised our Introduction to clarify the novelty of the present study (Page 2, lines 61-66). To our knowledge, this is the first study to examine the effects of timing of consecutive (i.e., 1 week) catechin ingestion on postprandial glucose metabolism in mice and humans. Our previous study reported on the effects of timing of acute (i.e., 1 time) catechin ingestion on postprandial glucose metabolism (Takahashi et al. Journal of Nutritional Biochemistry, 2019). Here, we hypothesized that the effectiveness of consecutive green tea intake on lowering glucose concentrations may differ depending on the time of day. In addition, some studies have reported that short term dietary interventions (from 4 to 6 days) could change glucose metabolism and circadian rhythm in humans (Jamshed et al. Nutrients, 2019; Wehrens et al. Current Biology, 2017). Moreover, it has been shown that consecutive intake of caffeine or catechin could modify the peripheral circadian rhythm in mice (Mi et al. Biochem Biophys Acta Mol Basis Dis, 2017; Narishige et al. Br J Pharmacol, 2014). We concluded no significant catechin intervention effects on glucose and insulin concentrations. Thus, we recommend that long-term research will be undertaken to confirm the chronic (i.e. 4-16 weeks) effects of catechin intake at different times on postprandial glucose in humans, and this has been added in the Discussion section (Page 13, lines 419-422).
I also have some concerns about the experimental part of the manuscript.
Query 1: Figure 6 showed that placebo groups also has detectable level of catechin. This raises the question whether the quantification of catechin is reliable or not. The authors did not provide details about how these data were obtained in the manuscript.
Response 1: Thank you for your attention to detail. We have revised the Figure 6 legend and made the following changes: “MP (Baseline)” to “MGT (Baseline)” and “EP (Baseline)” to “EGT (Baseline)”. In fact, in the MP and EP groups, in which catechin-rich green tea was not consumed, blood catechins were barely detected. We have also added the information on catechin concentration measurement (Page 6, lines 216-219).
Query 2: In Figure 6E, the error bars are missing.
Response 2: Thank you for spotting our mistake. We have added the error bars in Figure 6E as you suggested.
Query 3: The data showing statistically significant difference should be marked with “*” in all the figures. Please check Figures 6-8 and supplementary Figures S1-S3.
Response 3: In the original manuscript, we have described the results based on main or interaction effects using two- or three-factor ANOVA (i.e., trial, time, intervention effects). We think it might be confusing for the readers to interpret the data if we follow your suggestion. We have instead added the results for post-hoc tests in the revised manuscript when significant interaction effects were detected (Figures 6-7, supplementary Figures S1-S3, Page 9, lines 304-310, Page 10 lines 328-329).
Query 4: The data collected in section 2.2.6 and 2.2.7 were not presented in the results section. Due to the limited number of participates in the human study, I wonder whether the authors can get statistically significant results from these data or not.
Response 4: Regarding section 2.2.6, we have requested all participants to maintain their daily lifestyle (i.e., dietary and sleeping habits) and refrain doing certain activities, such as drinking alcohol, high-intensity exercise, which may influence our results. We have also presented the anthropometry and chronotype data in Table 2 and Figure S1(B). From these results, there were no significant effects of beverages on anthropometry and chronotype. Thus, our results involving postprandial glucose and insulin concentrations were mainly caused by the differences in timing (i.e., morning or evening) of catechin beverage ingestion.
Query 5: There are a few typos in the manuscripts as well, including “standardization” on line 188, “concentration” on line 113 and 229.
Response 5: We appreciate your attention to detail. We have corrected these errors accordingly (Page 4, line 115, Page 6, line 195, Page 7, line 241).
Once again, we thank the editor and reviewer for reading our manuscript so thoroughly and providing such constructive feedback. Hopefully, the modifications we have made in the new version of the manuscript and our responses to the reviewer’ concerns will be sufficient to render the paper suitable for publication in Nutrients.

Reviewer 2 Report
This is an interesting manuscript that describes the effects of timing of acute and consecutive catechin ingestion on postprandial glucose metabolism in mice and humans.
1) The Introduction sounds comprehensible and there’s no fragmented information. Also, it gives the necessary keywords to enter the investigated field.
2) The Methods are detailed and represent a guideline for similar approaches. However more information would be beneficial to strengthen the manuscript:
Please provide information on sample size determination for both the mice and human studies.
More information is needed on how the amount of total catechins were determined for the green tea beverage intervention in both mice and humans. How does this amount of catechins in green tea translate into what humans typically consume daily.
Did the authors control for gender in the statistical analyses in the human trial?
Did the authors collect dietary intake data during the human trial?
3) The Results are clear.
4) In the Discussion/Conclusions, the importance of longer human clinical trials could be better described besides the obvious necessity. For instance, mid-to-long-term detection of glucose and insulin concentrations might disclose interesting aspects of specific oscillation around the setpoints that could reveal actual effects on metabolic shifts induced by this category of compounds.
Author Response
Paper No. Nutrients- 692047 R1
Response to Reviewers’ Comments
Once again, we wish to thank the editor and reviewers for reading our manuscript so thoroughly and providing such constructive feedback. The quality of our manuscript has certainly improved as a result of these comments. Our point-by-point responses are provided below, and the necessary changes are highlighted in yellow in the revised manuscript.
Reviewer 2:
General Comments
This is an interesting manuscript that describes the effects of timing of acute and consecutive catechin ingestion on postprandial glucose metabolism in mice and humans.
The Methods are detailed and represent a guideline for similar approaches. However, more information would be beneficial to strengthen the manuscript:
Query 1: Please provide information on sample size determination for both the mice and human studies.
Response 1: As you suggested, we have added the information on sample size determination in human experiments (Page 6-7, lines 231-233). Based on the distribution of postprandial glucose values in our previous study, the sample size was calculated to be able to detect a large effect (Cohen's d=0.98). A calculated sample size of 11 was required to approximately have 80% power to detect large effects at a significance level of 0.05. In our study, the sample size in each beverage group (placebo group = 19, catechin group = 19) and intake timing (morning group = 19, evening group = 19) met the above criteria.
Regarding animal experiments, we estimated the sample sizes in each group (n = 6) based on previous studies (Forester et al. Mol. Nutr Food Res. 2012). We have added this information accordingly (Page 2, lines 89-90, Page 3, line 107). From the perspective of animal welfare, the 3R (Replacement, Reduction, Refinements) principle has been recommended by several guidelines. Thus, we have refrained from performing additional experiments in mice once we reached significant differences and proceeded to perform human experiments.
Query 2: More information is needed on how the amount of total catechins were determined for the green tea beverage intervention in both mice and humans. How does this amount of catechins in green tea translate into what humans typically consume daily.
Response 2: Thank you for your thoughtful comment. We have determined the amount of total EGCG and catechin based on previous studies (Animal experiments: Forester et al. Mol. Nutr Food Res. 2012; human experiments: Takahashi et al. British Journal of Nutrition, 2014, Takahashi et al. Journal of Nutritional Biochemistry, 2019). From these studies, the specific amount of EGCG or catechin we used in the present study resulted in decreased postprandial glucose concentrations in both mice and human. This amount is equivalent to 5-6 cups of green tea servings daily (Page 2, lines 94-95, Page 5, lines 187-189).
Query 3: Did the authors control for gender in the statistical analyses in the human trial?
Response 3: Yes, we conducted sub-analysis to consider gender. However, we found no significant differences as sample size was too small to detect gender difference. In the manuscript, we have shown the total of number of men and women participants.
Query 3: Did the authors collect dietary intake data during the human trial?
Response 3: We did not collect information about the daily dietary intake during the experimental period. Instead we have requested all participants to maintain their daily lifestyle (i.e., dietary and sleeping habits) and refrain doing certain activities, such as drinking alcohol, high-intensity exercise, which may influence our results. From our findings, there were no significant changes in body weight and fasting metabolic parameters. Thus, the effects of the different dietary habits of participants on glucose metabolism were minimised.
Query 4: In the Discussion/Conclusions, the importance of longer human clinical trials could be better described besides the obvious necessity. For instance, mid-to-long-term detection of glucose and insulin concentrations might disclose interesting aspects of specific oscillation around the setpoints that could reveal actual effects on metabolic shifts induced by this category of compounds.
Response 4: Thank you for your suggestion. We will take your suggestion into consideration in the future research.
Once again, we thank the editor and reviewer for reading our manuscript so thoroughly and providing such constructive feedback. Hopefully, the modifications we have made in the new version of the manuscript and our responses to the reviewer’ concerns will be sufficient to render the paper suitable for publication in Nutrients.

Reviewer 3 Report
I would like to congratulate authors on this interesting study. Please consider my comments:
line 42; I do not agree; elevated postprandial glucose concentration are higher in the morning, after the breakfast which is connected with hormons; papers 6-8 cited in paper are niche the examined groups are small EGCG is very sensitive substance... the temperature of water should be pointed as well as time between preparing the beverages and consumption please try to explain why the EGCG do not improove the insulin impact but only stimulate its secretionAuthor Response
Paper No. Nutrients- 692047 R1
Response to Reviewers’ Comments
Once again, we wish to thank the editor and reviewers for reading our manuscript so thoroughly and providing such constructive feedback. The quality of our manuscript has certainly improved as a result of these comments. Our point-by-point responses are provided below, and the necessary changes are highlighted in yellow in the revised manuscript.
Reviewer 3:
General comments
I would like to congratulate authors on this interesting study. Please consider my comments:
Query 1: line 42; I do not agree; elevated postprandial glucose concentration are higher in the morning, after the breakfast which is connected with hormons; papers 6-8 cited in paper are niche the examined groups are small EGCG is very sensitive substance... the temperature of water should be pointed as well as time between preparing the beverages and consumption please try to explain why the EGCG do not improove the insulin impact but only stimulate its secretion
Response 1: We apologize for the confusion. Lines 41-43 describes the relationship between timing diet and postprandial glucose metabolism and not between catechin intake and postprandial glucose metabolism. Although mechanisms for stimulating insulin secretion by EGCG intake in the evening are unclear, it might be related to time-of-day variations in insulin receptors and insulin receptor substrate-1 which stimulates glucose uptake. Addition research is required to clarify this molecular mechanism.
Aside from that, we have provided the temperature of water for each beverage (Page 5, line 178). We have served the placebo and catechin beverages at room temperature (20–25℃) to minimize the effects of beverage temperature on our results.
Once again, we thank the editor and reviewer for reading our manuscript so thoroughly and providing such constructive feedback. Hopefully, the modifications we have made in the new version of the manuscript and our responses to the reviewer’ concerns will be sufficient to render the paper suitable for publication in Nutrients.

Round 2
Reviewer 1 Report
I do not believe the manuscript has been significantly improved and should not warrants publication in Nutrients. I maintain my opinion that although it is a sound manuscript, there is limited originality in the manuscript.